# Effect of Hydrogen on Corrosion Behavior of 321 Stainless Steel in NH_4_Cl Solution

**DOI:** 10.3390/ma15197010

**Published:** 2022-10-10

**Authors:** Xiuqing Xu, Wei Wang, Tao Suo, Lei Jiang, Xiangkun Yin, Guangxu Cheng, Anqing Fu, Fang Yang

**Affiliations:** 1State Key Laboratory of Performance and Structural Safety for Petroleum Tubular Goods and Equipment Materials, R&D Center of TGRI, Xi’an 710077, China; 2School of Chemical Engineering and Technology, Xi’an Jiaotong University, Xi’an 710049, China; 3Petrochina Changqing Petrochemical Company, Xianyang 712042, China; 4Petrochina Dalian Petrochemical Company, Dalian 116031, China

**Keywords:** 321 stainless steel, electrolytic hydrogen charging, NH_4_Cl solution, corrosion behavior

## Abstract

For a hydrogenation heat exchanger operating under severe working conditions such as high temperature, high pressure and a hydrogen environment, perforation accidents caused by NH_4_Cl corrosion occur frequently. However, few reports on the effect of hydrogen on the corrosion behavior of metal materials in NH_4_Cl aqueous solution have been published. In this paper, X-ray photoelectron spectroscopy (XPS), electrochemical dynamic potential polarization, electrochemical impedance spectroscopy (EIS), Mott–Schottky (M-S) curves and scanning electron microscopy (SEM) were used to study the effect of electrochemical hydrogen charging (EHC) on the corrosion behavior of 321 stainless steel in an NH_4_Cl solution environment. The results show that: (1) hydrogen can change the structure and chemical composition of 321 stainless steel passive film and promote the conversion of metal oxide to hydroxide. At the same time, it can reduce the stability of the passive film. (2) Hydrogen can increase the thermodynamic and kinetic tendency of corrosion reaction and cooperate with Cl^−^ to promote the occurrence of pitting corrosion.

## 1. Introduction

As the core equipment of hydrogenation units, the safe operation of hydrogenation heat exchangers is very important to the development of China’s petrochemical industry. Since hydrogenation heat exchangers operate under severe working conditions such as high temperature, high pressure and hydrogen for a long time, accidents caused by NH_4_Cl corrosion and hydrogen embrittlement frequently occur [1,2,3].

The service environment of a hydrogenation heat exchanger is complex, including hydrogen, stress and a corrosive medium (mainly ammonium salt). The coupling of various factors can cause many material damages, such as stress corrosion and hydrogen embrittlement [4,5,6]. However, few reports on the effect of hydrogen on the corrosion behavior of metal materials in NH_4_Cl aqueous solution have been published. Ou et al. [7] analyzed the movement and deposition mechanism of ammonium salt in a heat exchanger through the coupling of the flow field, temperature field and concentration field. Hai-bo Wang et al. [8] investigated the effect of different factors, including surface roughness, temperature, dissolved oxygen, pH and H_2_S concentration, on the corrosion behavior of carbon steel in an NH_4_Cl environment. Toba et al. [9] investigated the corrosion behavior of carbon steel and alloys in high-concentration NH_4_Cl solutions.

Due to the frequent failures of hydrogenation heat exchangers, material upgrading and water injection have been adopted as the main methods to control corrosion [10,11,12]. At present, the materials of heat exchanger tubes are gradually upgraded from carbon steel and stainless steel to Inconel 625 alloys and Incoloy 825 alloys [13,14,15]. However, nickel-based alloys are rarely used due to their high price. The main material of heat exchangers is 15CrMo steel and 321 stainless steel. Therefore, studying the typical corrosion mechanism of hydrogenation heat exchangers in multi-factor service environments is of great significance to safety during production and economic saving. In this paper, a typical AISI 321 stainless steel is selected to study the corrosion mechanism of hydrogen on 321 stainless steel in NH_4_Cl and hydrogen environments.

## 2. Materials and Methods

### 2.1. Materials

Here, 321 austenitic stainless steel with a size of 10 mm × 10 mm × 3 mm was used as an experimental material; its chemical composition is shown in Table 1. The microstructure was characterized by a metallographic microscope (MEF4M, Heidelberg, Germany, PLEA LEICA), as shown in Figure 1.

### 2.2. Electrochemical Hydrogen Charging

Electrochemical hydrogen charging was carried out at the temperature of 45 ± 1 °C, and the hydrogen charging solution was 0.5 mol/L H_2_SO_4_ + 1.85 mol/L Na_4_P_2_O_7_. The sample and Pt electrodes were used as the working electrode (WE) and counter electrode (CE), respectively. The reference electrode (RE) was a saturated Ag/AgCl electrode, as shown in Figure 2. Analytical reagent-grade chemicals and distilled water were used to prepare the solution. The hydrogen charging time was 180 min, and the hydrogen charging current densities were 0.1 mA/cm^2^, 1 mA/cm^2^, 10 mA/cm^2^ and 80 mA/cm^2,^ respectively. The sample was ultrasonically cleaned immediately with deionized water and alcohol after hydrogen charging. At the same time, the time interval between electrochemical hydrogen charging and the subsequent electrochemical test should be controlled within 10 min to prevent the overflow of hydrogen.

### 2.3. Electrochemical Test

A C350 electrochemical workstation (manufactured by Wuhan CorrTest Instrument Co. Ltd., Wuhan, China) was used to test electrochemical behaviors, and the data were obtained in a three-electrode mode. The reference electrode was a saturated Ag/AgCl electrode (SCE), and the auxiliary electrode was a graphite electrode. The electrolyte was 1 wt% NH_4_Cl aqueous solution. Nitrogen was continuously injected into the solution during the test, and the test temperature was controlled at 45 ± 1 °C.

To study the effect of hydrogen on the corrosion behavior in NH_4_Cl solution, a potentiodynamic polarization test and electrochemical impedance spectroscopy (EIS) test were performed. The specimen was allowed to stabilize at open circuit potential (OCP) for 60 min at room temperature before the potentiodynamic polarization test and EIS test. Each test was repeated at least three times to ensure reproducibility. The range of potential for the potentiodynamic polarization test was from −300 mV to 1500 mV at a constant scan rate of 1 mV/s. The frequency range of the EIS measurement was from 0.02 Hz to 10^5^ Hz.

### 2.4. Characterization

In order to study the effect of hydrogen on the semiconductor properties and electronic structure of the passive film, the capacitance and potential were measured, and the results were analyzed by an M-S curve. Before the test, the sample was passivated for 3 h under open circuit conditions. The scanning voltage range was from—1.0 V to 1.0 V, and the scanning rate and frequency were 20 mV/s and 1 kHz, respectively. The surface morphology of the samples was observed by scanning electron microscopy (SEM, MAIA3LMH). The chemical composition of the passive film on the surface of 321 stainless steel was studied by X-ray photoelectron spectroscopy (XPS, Thermo Fisher ESCALAB Xi +, Shanghai, China) with Al Kα radiation (1486.6 eV). The incident angle was 45°, and the test depth was about 10 nm.

## 3. Results and Discussion

### 3.1. Effect of Hydrogen on Passive Film Structure of 321 Stainless Steel

Figure 3 shows the XPS spectrums for Cr 2p3/2, Fe 2p3/2, and O 1s regions of passive film on the surface of 321 stainless steel before and after EHC (1 mA/cm^2^). Table 2 lists the XPS peak positions and peak areas of different components of the passive film.

As seen, Fe elements mainly exist in the form of Fe_2_O_3_, FeOOH, FeO and a little Fe_3_O_4_ in the passive film before EHC. The peak of Fe_2_O_3_ decreases and its content decreases from 59.66% to 28.96% after EHC for 3 h. At the same time, the contents of Fe_3_O_4_ and FeO increase from 5.19% and 17.00% to 9.17% and 33.28%, respectively. Obviously, hydrogen charging reduces the Fe^3+^ content and increases the Fe^2+^ content in the passivation film. The difference in the radius of Fe^3+^ and Fe^2+^ will lead to lattice expansion. Therefore, the stability of the passivation film will reduce under the synergistic action of corrosion and hydrogen [16,17].

Compared with the contents of Cr element in passive film before EHC, the content of Cr metal and Cr(OH)_3_ in the passivation film increases from 7.43% and 39.32% to 11.79% and 44.01%, respectively. On the contrary, the content of Cr_2_O_3_ decreases from 53.25% to 44.2% after EHC. As we know, Cr_2_O_3_ is in a more stable state than Cr(OH)_3_, which can better protect the metal matrix. This result indicates that the protection of the passive film is reduced after hydrogen charging. Similarly, the content of O^2−^ and OH^−^ in the passivation film decreases from 13.61% and 69.33% to 8.03% and 59.32%, respectively. Additionally, the content of H_2_O increases from 17.06% to 32.65% after EHC. The ratio of O^2−^/OH^−^ also decreases from 0.2 to 0.14, indicating that hydrogen promoted the transformation from O^2−^ to OH^−^, which reduces the stability of the passivation film and induces pitting corrosion.

### 3.2. Effect of Hydrogen on Electrochemical Properties of 321 Stainless Steel

Figure 4 shows the Nyquist and Bode plots of samples with different hydrogen charging current densities (different hydrogen concentrations) in 1 wt% NH_4_Cl aqueous solution. The results show that the diameter of the depressed semi-circle obviously decreases with the increasing hydrogen charging current density from 0 mA/cm^2^ to 80 mA/cm^2^, which suggests that hydrogen decreases the protection of the passive film.

Fitting the experimental results with the ZSimpWin software V3.6 (Wuhan, China), the equivalent circuit was obtained, as shown in Figure 5. Table 3 displays the EIS analysis results under different hydrogen charging current densities in 1 wt% NH_4_Cl aqueous solution by means of the equivalent circuit. As it can be seen, the difference in the resistance of NH_4_Cl solution (*R_s_*) under different hydrogen charging current densities is small. The value of the passive film impedance (*R_f_*) without EHC is higher than that under the condition of EHC. *R_f_* and the charge transfer impedance *R_ct_* reduce from 1.39 × 10^4^ Ω·cm^2^ and 8.20 × 10^4^ Ω·cm^2^ to 1.42 × 10^3^ Ω·cm^2^ and 2.04 × 10^3^ Ω·cm^2,^ respectively, as the hydrogen charging current density gradually increases from 0 mA/cm^2^ to 80 mA/cm^2^. The decrease in *R_f_* can be attributed to the thinning of the passive film, the reduction of Fe^3+^ to Fe^2+^ and the decrease in metal oxide content. The decrease in *R_ct_* after hydrogen charging is due to the introduction of active sites in the H oxidation reaction, which enhances the electrochemical activity of the passive film.

Generally, the polarization resistance *R_p_* (*R_p_* = *R_f_* + *R_ct_*) is used to illustrate the corrosion resistance of metals, and the corrosion rate is inversely proportional to the polarization resistance [5]. The calculated *R_p_* of the sample without EHC is 9.86 times and 27.72 times that under the hydrogen charging current density of 10 mA/cm^2^ and 80 mA/cm^2,^ respectively, which indicates that hydrogen can significantly reduce the corrosion resistance of the passive film.

Figure 6 shows the polarization curve of 321 stainless steel with different hydrogen charging current densities in 1 wt% NH_4_Cl aqueous solution. Their corrosion current densities *I_corr_* are calculated by means of the Tafel extrapolation method. Meanwhile, the obtained results from the polarization curve test under different conditions results are listed in Table 4. As seen, there are passivation zones in the polarization curves at hydrogen charging current densities of 0.1 mA/cm^2^ and 1 mA/cm^2^ that are similar to that without EHC. However, the pitting potential *E_pit_* decreases and the dimensional passivation current density *I_pass_* increases. When the hydrogen charging current increases to 10 mA/cm^2^ and 80 mA/cm^2^, there is no obvious passivation zone, indicating that hydrogen can change the polarization behavior of the sample surface, and the pitting potential decreases sharply. The above results show that hydrogen increases the pitting corrosion sensitivity of 321 stainless steel, and the pitting corrosion resistance decreases with the increase in hydrogen charging current density.

As observed in Figure 7, the corrosion potential *E_corr_* of all hydrogen-filled samples decreased significantly, and the *I_corr_* increased gradually compared with the samples without EHC. Meanwhile, the *E_corr_* of 321 stainless steel become negative from −0.131 V to −0.291 V, and *I_corr_* increases from 0.593 μA·cm^−2^ to 17.418 μA·cm^−2^ when the hydrogen charging current density is from 0 mA/cm^2^ to 80 mA/cm^2^. This result illustrates that hydrogen can significantly reduce the corrosion resistance of 321 stainless steel. On the one hand, the pre-charging hydrogen makes the austenite structure of 321 stainless steel transform to martensite [18], which accelerates the corrosion process of 321 metal matrix under galvanic corrosion. On the other hand, hydrogen causes micro-defects such as vacancies, dislocations and interface damage, resulting in hydrogen softening, which makes the sample more prone to anodic dissolution and corrosion in the corrosive medium.

### 3.3. Corrosion Morphology Analysis

Figure 8 shows the corrosion morphology of 321 stainless steel after an electrochemical test under different hydrogen charging current densities. As observed, the number and size of pits generally increase with the increase in hydrogen charging current density. When the hydrogen charging current density is from 0 mA/cm^2^ to 1 mA/cm^2^, it can be seen that there is an obvious cover on the pit, which is considered to be the residue of the passive film [19,20]. The covering as a barrier for ion diffusion hinders the mass transfer process inside the pit, which helps to form a “small and deep” pitting structure. It is a typical structure of pitting in passivation systems [21], as seen in Figure 8a. However, the structure and chemical composition of the passive film change greatly and the stability decreases (Figure 3) when the hydrogen charging current density is from 10 mA/cm^2^ to 80 mA/cm^2^, which weakens the protective ability of the passive film. At the same time, hydrogen strengthens the corrosion of chloride ions and accelerates the anodic dissolution of the passive film (Figure 6). Therefore, the covering disappears, and the sample surface presents a “large and shallow” corrosion structure, as seen in Figure 8d,e. This is also consistent with the result that there is no obvious passivation zone in the polarization curve at this hydrogen charging current density.

### 3.4. Effect of Hydrogen on Semiconductor Characteristics of Passive Film

Figure 9 shows the M-S curve of 321 stainless steel under different hydrogen charging current densities in 1 wt% NH_4_Cl aqueous solution. As seen, there are three approximate linear regions with different slopes in the M-S curve, which represent different semiconductor characteristics. The potential interval of Part 1 and Part 3 shows P-type semiconductor characteristics [22], and Part 2 shows N-type semiconductor characteristics. The chemical composition and structure of the passivation film determine the semiconductor characteristics of the passivation film. It is generally considered that the inner layer of the passivation film is a P-type semiconductor that is composed of oxides of Cr, Ni or Mo and oxygen vacancies. The outer layer is an N-type semiconductor that is composed of Fe oxide or hydroxide [23]. Combined with the XPS test results, it can be concluded that the outer passivation film is composed of Fe_3_O_4_, Fe_2_O_3_ and FeOOH. The main component of the inner layer is Cr_2_O_3_, which constitutes a P-type semiconductor.

The bipolar N-P semiconductor structure indicates that the passive film can prevent the ions from passing between the solution and the metal matrix to improve the corrosion resistance of stainless steel. With the increase in hydrogen charging current density, the slopes of the three-section curves decrease, and the bipolar characteristics disappear gradually. This phenomenon shows that the concentration of metal ion vacancy and oxygen vacancy in the passivation film increases after hydrogen charging. Fe^3+^ reacts with H to form H^+^ and Fe^2+^, and H^+^ reacts with O^2−^ in the passive film to form oxygen vacancy and OH^−^. The combination of Cl^−^ and oxygen vacancy promotes the Mott–Schottky pair reaction, which increases the concentration of oxygen vacancy and metal ion vacancy in the passive film. The generated oxygen vacancy can continue to react with Cl^−^. Metal ion vacancies gather at the interface resulting in the separation of the metal matrix and the passive film. Thus it breaks the dynamic balance between the dissolution and growth of the passive film, finally leading to the penetrating rupture of the passive film.

## 4. Conclusions

The influence of electrochemical charging on the corrosion behavior of 321 stainless steel in NH4Cl solution was systematically studied using XPS, electrochemical potentiodynamic polarization, EIS, M-S curve and SEM. The main conclusions are as follows:(1)Hydrogen can change the structure and chemical composition of a passivation film on 321 stainless steel and promote the reduction of Fe^3+^ to Fe^2+^ and the conversion of O^−-^ to OH^−^ and H_2_O. At the same time, it reduces the stability of the passivation film.(2)The polarization resistance decreases with the increase in hydrogen charging current density, which improves the corrosion rate of the metal. Hydrogen can increase the thermodynamic and kinetic tendency of corrosion reaction and cooperate with Cl^−^ to promote the occurrence of pitting corrosion.(3)Under high hydrogen concentrations (10−80 mA/cm^2^), hydrogen can promote the transformation of the pitting structure of 321 stainless steel from “small and deep” to “large and shallow”.f Hydrogen reduces the protective performance of passive film and inhibits the formation of the passive film.

## Figures and Tables

**Figure 1 materials-15-07010-f001:**
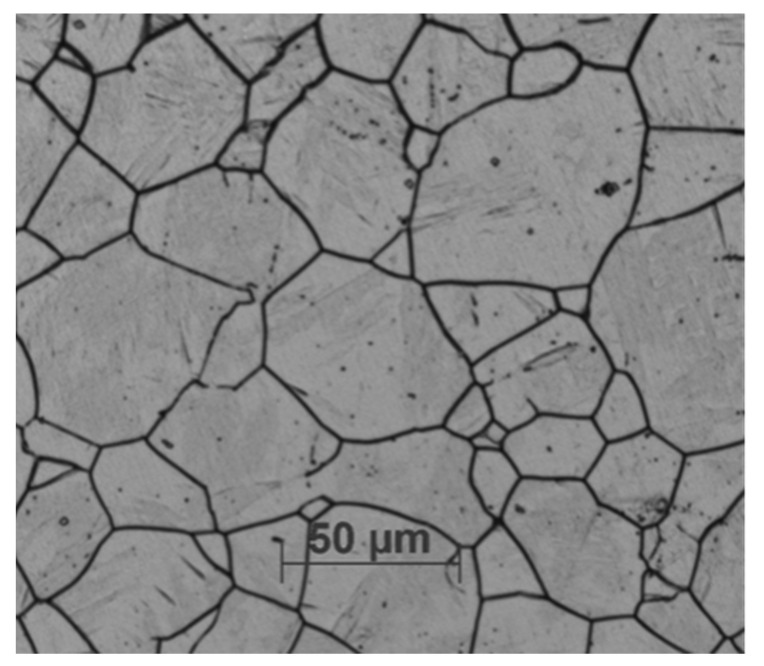
The microstructure of 321 austenitic stainless steel.

**Figure 2 materials-15-07010-f002:**
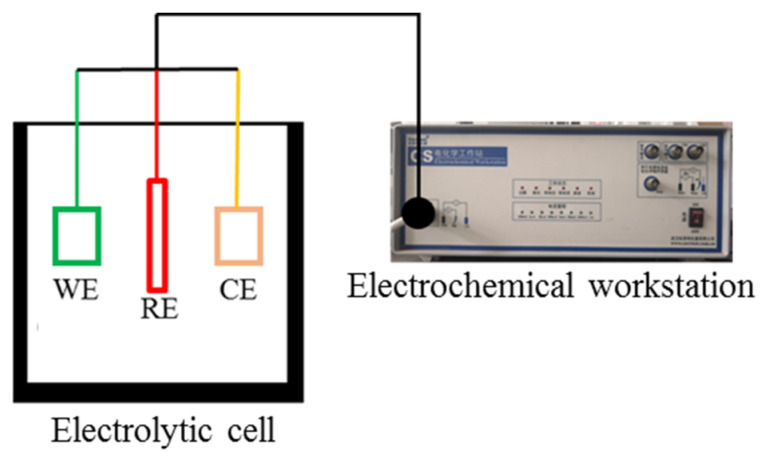
Schematic diagram of electrochemical hydrogen charging equipment.

**Figure 3 materials-15-07010-f003:**
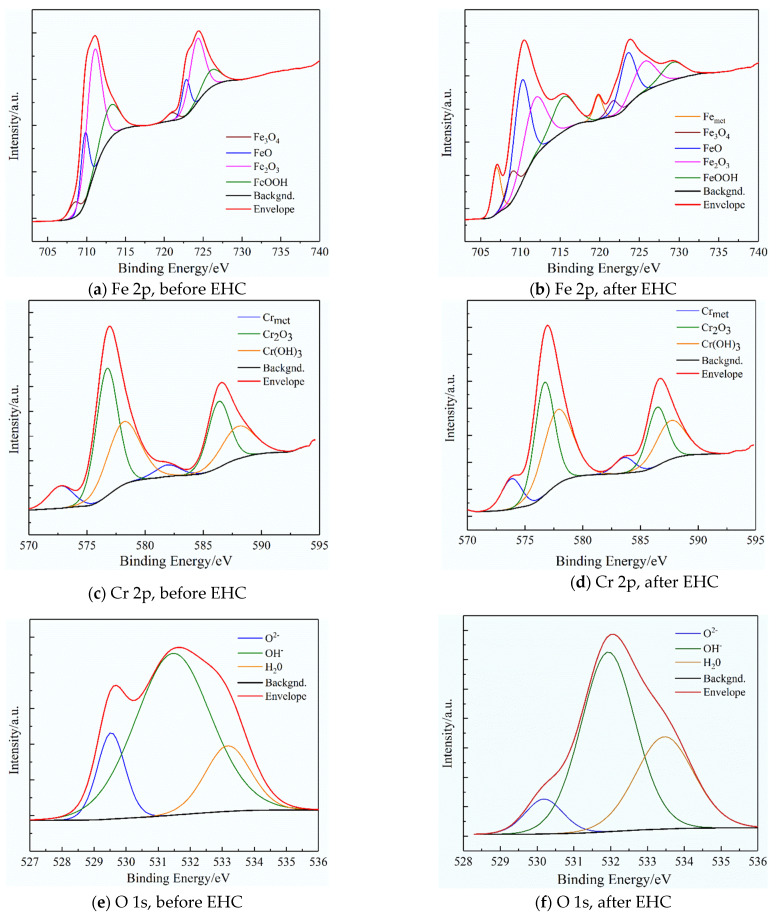
XPS spectra for passive film on the surface of 321 stainless steel before and after EHC (1 mA/cm^2^).

**Figure 4 materials-15-07010-f004:**
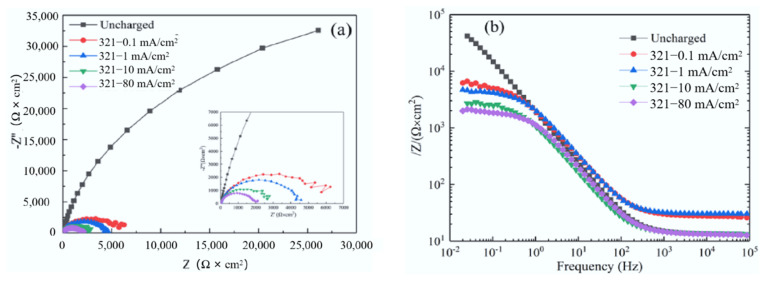
(**a**) Nyquist plots for different hydrogen charging current densities regarding the proposed test condition and (**b**) corresponding impedance results.

**Figure 5 materials-15-07010-f005:**
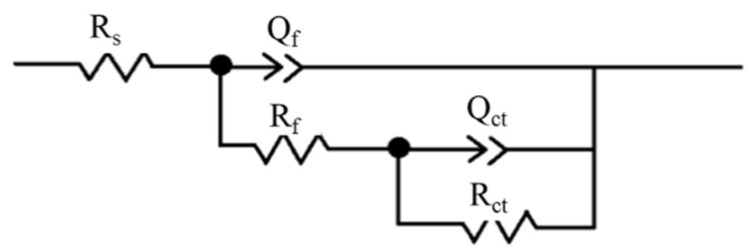
Equivalent circuits used for modelling the EIS results.

**Figure 6 materials-15-07010-f006:**
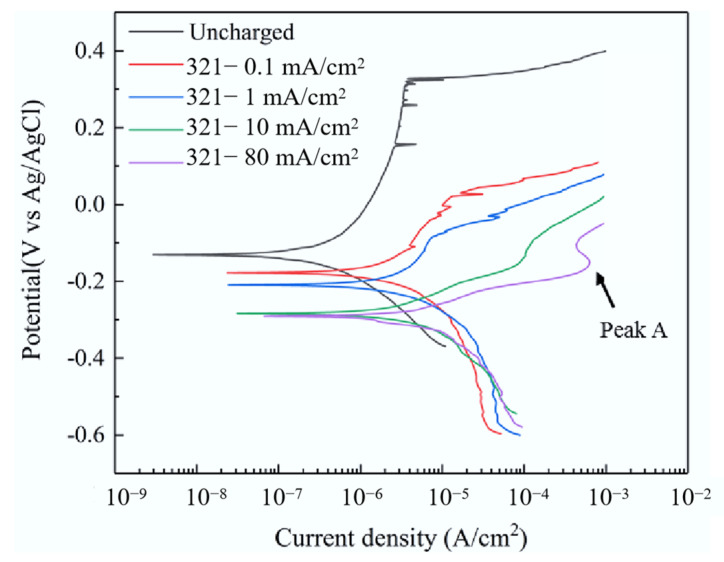
The polarization curve of 321 stainless steel with different hydrogen charging current densities in 1 wt% NH_4_Cl aqueous solution.

**Figure 7 materials-15-07010-f007:**
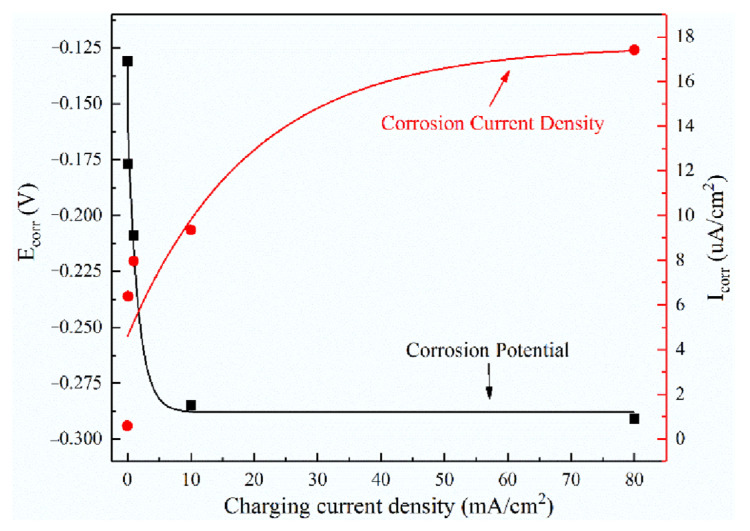
Electrochemical parameters of Tafel polarization curves.

**Figure 8 materials-15-07010-f008:**
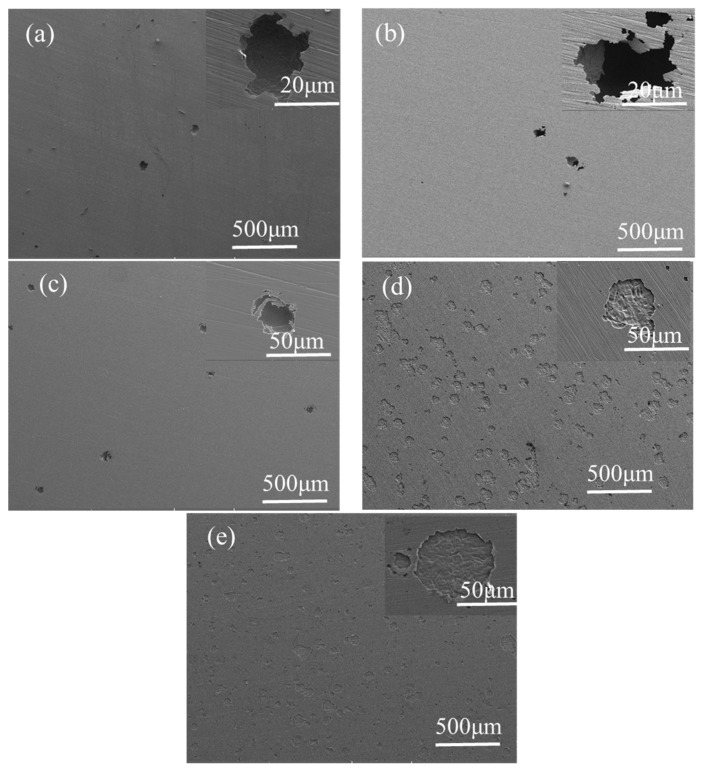
SEM micrographs of 321 stainless steel after polarization under the condition of (**a**) without EHC and a hydrogen charging current density of (**b**) 0.1 mA/cm^2^, (**c**) 1 mA/cm^2^, (**d**) 10 mA/cm^2^, (**e**) and 80 mA/cm^2^.

**Figure 9 materials-15-07010-f009:**
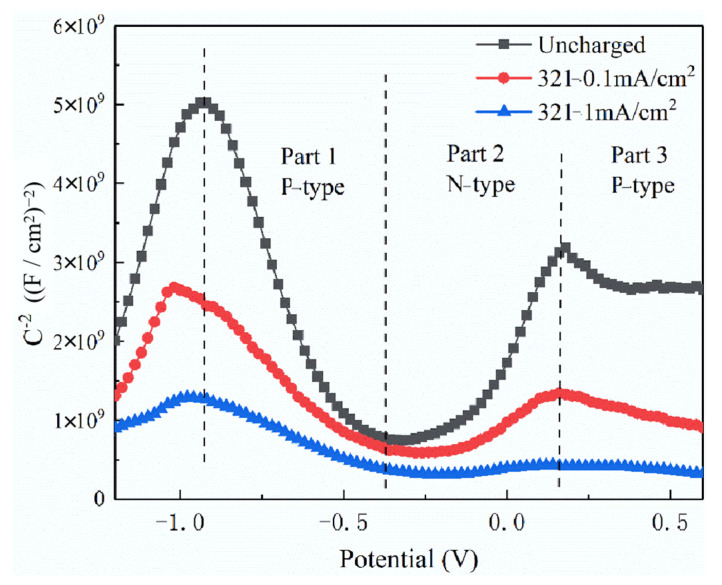
The M-S curve of 321 stainless steel under different hydrogen charging current densities in 1 wt% NH_4_Cl aqueous solution.

**Table 1 materials-15-07010-t001:** Chemical composition of 321 stainless steel (wt %).

C	Si	Mn	S	Ni	Cr	Ti	Fe
0.02	0.63	2.23	0.70	10.03	18.05	0.28	Bal.

**Table 2 materials-15-07010-t002:** The XPS peak positions and peak areas of different components of passive film.

Spectrum	Component	Binding Energy/eV	Peak Area /%	
			before EHC	after EHC
Fe 2p3/2	Fe	706.7	0	9.11
	Fe_3_O_4_	708.8	5.19	9.17
	FeO	709.6	17	33.28
	Fe_2_O_3_	710.8	59.66	28.96
	FeOOH	711.4	18.15	19.48
Cr 2p3/2	Cr	574.1	7.43	11.79
	Cr_2_O_3_	576.5	53.25	44.20
	Cr(OH)_3_	577.4	39.32	44.01
O 1s	O^2−^	530.2	13.61	8.03
	OH^−^	531.3	69.33	59.32
	H_2_O	533.2	17.06	32.65

**Table 3 materials-15-07010-t003:** Impedance parameters under different hydrogen charging current densities.

Condition	*R_s_*/Ω·cm^2^	*Q_f_*/F·cm^−2^	*R_f_*/Ω·cm^2^	*Q_ct_*/F·cm^−2^	*R_ct_*/Ω·cm^2^
Without EHC	13.0	2.03 × 10^−5^	1.39 × 10^4^	9.45 × 10^−5^	8.20 × 10^4^
0.1 mA/cm^2^	14.4	3.46 × 10^−5^	2.98 × 10^3^	7.89 × 10^−5^	6.75 × 10^3^
1 mA/cm^2^	10.0	1.51 × 10^−5^	3.05 × 10^3^	8.02 × 10^−5^	4.43 × 10^3^
10 mA/cm^2^	10.2	5.02 × 10^−5^	1.38 × 10^3^	1.64 × 10^−4^	2.76 × 10^3^
80 mA/cm^2^	10.0	3.77 × 10^−5^	1.42 × 10^3^	1.11 × 10^−4^	2.04 × 10^3^

**Table 4 materials-15-07010-t004:** The obtained results from polarization curve test under different conditions.

Condition	*E_corr_*/V	*I_corr_*/μA·cm^−2^	*E_pit_*/V	*I_pass_*/μA·cm^−2^
Without EHC	−0.131	0.593	0.315	3.612
0.1 mA/cm^2^	−0.177	6.384	0.021	12.884
1 mA/cm^2^	−0.209	7.965	−0.084	7.517
10 mA/cm^2^	−0.285	9.349	/	/
80 mA/cm^2^	−0.291	17.418	/	/

## Data Availability

Not applicable.

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
