# Peer review of "Effect of Hydrogen on Corrosion Behavior of 321 Stainless Steel in NH4Cl Solution"

_materials, 2022, doi:10.3390/ma15197010_

Round 1

Reviewer 1 Report

The presented manuscript seems to be interesting for readers of the Materials journal, it is written in a good manner and suits the requirements of the journal. It can be accepted for publication after minor corrections listed below.

- English language of manuscript is acceptable in general. However, it would be much better to improve. Please avoid the unnecessary long sentence. Also, some grammatical and typos mistakes can be observed. For example: chemicalsand,

- Acronyms, such as M-S should all be defined at their first occurrence in the manuscript;

- The authors stated in the "Materials and methods" section that: ”The hydrogen charging time is 180 min, and the hydrogen charging current density is 0.1, 1, 10 and 80 mA/cm2 respectively.”  It seems that the sentence should be revised. Also, the following sentence is not clear and should be rewritten:” At the same time, the electrochemical hydrogen charging time should be controlled within 10 min to prevent the overflow of H.”

- In the "Materials and methods" section, the details of the materials (grain size, structure of 321 stainless steel, production method and heat treatment before corrosion testing, etc.) and the method (auxiliary electrode, reference electrode, etc.) are not fully explained. Also, based on the description of the text, Figure 1 is not complete and should be modified or can be deleted

- The details of the work method in the section "1.2. Electrochemical hydrogen charging" and "1.3. Electrochemical test" are not understandable and should be rewritten.

- It is suggested to rewrite the following text:” The scanning voltage range is - 1.0V - 1.0V”

- The authors state in the section that: “And Rf and the charge transfer impedance Rct reduce from 1.39×104 Ω·cm2 and 8.20×104 Ω·cm2 to 1.42×103 Ω·cm2 and 2.03×103 Ω·cm2 respectively as the hydrogen charging current density gradually increases from 0 mA/ cm2 to 80 mA/cm2.” 2.03x103 should be corrected to 2.04x103. It is suggested not to start the sentence with and.

- Literature review is not sufficient and authors must review and cite more papers in the field and especially newly published ones. Doing this, review and citing the following refs could be helpful:

[] Journal of Materials Engineering and Performance, 27, 2018, 271-281.

[] International Journal of Pressure Vessels and Piping, 145, 2016, 1-12

[] Measurement, 75, 2015, 5-11.

Author Response

Thank you for your comments and suggestions on our manuscript which are very useful for the improvement of our manuscript. We have revised the manuscript according to the reviewers’ suggestions. All the changes are color-marked in the revised manuscript. The following are the detailed corrections and revisions point by point:

(1)  English language of manuscript is acceptable in general. However, it would be much better to improve. Please avoid the unnecessary long sentence. Also, some grammatical and typos mistakes can be observed. For example: chemicalsand,.

Answer: According to the reviewers’ comment, we have corrected the language carefully and hope this revised manuscript will be convincing.

(2)   Acronyms, such as M-S should all be defined at their first occurrence in the manuscript.

Answer: Thank the reviewers for the comment. The M-S is defined in Abstract in revised manuscript.

(3) The authors stated in the "Materials and methods" section that: “The hydrogen charging time is 180 min, and the hydrogen charging current density is 0.1, 1, 10 and 80 mA/cm2 respectively.”  It seems that the sentence should be revised. Also, the following sentence is not clear and should be rewritten:” At the same time, the electrochemical hydrogen charging time should be controlled within 10 min to prevent the overflow of H”.

Answer: Your comment is greatly appreciated. According to the reviewers’ suggestion, Part 1.2 in "Materials and methods" is rewritten as seen in revised manuscript.

(4) In the "Materials and methods" section, the details of the materials (grain size, structure of 321 stainless steel, production method and heat treatment before corrosion testing, etc.) and the method (auxiliary electrode, reference electrode, etc.) are not fully explained. Also, based on the description of the text, Figure 1 is not complete and should be modified or can be deleted.

Answer: Thanks for reviewers’ comment. The details of materials and Figure 1 are modified in Part 1.1 and Part 1.2 in "Materials and methods" respectively.

  (5) The details of the work method in the section "1.2. Electrochemical hydrogen charging" and "1.3. Electrochemical test" are not understandable and should be rewritten.

Answer: According to the reviewers’ comment, we rewritte the section "1.2. Electrochemical hydrogen charging" and "1.3. Electrochemical test", seen in revised manuscript. 

(6) It is suggested to rewrite the following text:” The scanning voltage range is - 1.0V - 1.0V”.

Answer: According to the reviewers’ comment, this sentence is rewritten as follows:” The scanning voltage range is from - 1.0V to 1.0V”.

(7) The authors state in the section that: “And Rf and the charge transfer impedance Rct reduce from 1.39×104 Ω·cm2 and 8.20×104 Ω·cm2 to 1.42×103 Ω·cm2 and 2.03×103 Ω·cm2 respectively as the hydrogen charging current density gradually increases from 0 mA/ cm2 to 80 mA/cm2.” 2.03×103 should be corrected to 2.04×103. It is suggested not to start the sentence with and.

Answer: According to reviewer’s suggestion, this sentence is revised.

 (8)  Literature review is not sufficient and authors must review and cite more papers in the field and especially newly published ones. Doing this, review and citing the following refs could be helpful:[] Journal of Materials Engineering and Performance, 27, 2018, 271-281.[] International Journal of Pressure Vessels and Piping, 145, 2016, 1-12[] Measurement, 75, 2015, 5-11.

Answer: Thanks for the reviewer’s comment. The References are expanded and reworked in revised manuscript.

Reviewer 2 Report

1. English needs to be further improved. Some sentences are not even completed, here are just a couple of examples:

"However, little report on the effect of hydrogen on the corrosion behavior of metal materials in NH4Cl aqueous solution."

"To study the effect of hydrogen on the corrosion behavior and passive film in NH4Cl solution, the potentiodynamic polarization curve and Electrochemical Impedance Spectroscopy (EIS) were performed." 

2. What exactly are the composition and thickness of the passivation layer on the 321 SS? Can the author comment on it based on XPS measurements and/or literature reports?

3. Based on the very comprehensive analyses in the manuscript, I agree with the authors that the passivation film loses its protection as the charging of hydrogen increases. This agrees with many other published reports. However, the authors state that the charging of hydrogen makes the passivation film thinner. I do not see direct evidence from the manuscript. Did the authors measure (or estimate) the thicknesses of the passivation layer before and after EHC? Or, do the authors just mean "pitting corrosion" instead? I might misunderstand the statement if the authors can clarify that would be great. 

Author Response

Thank you for your comments and suggestions on our manuscript which are very useful for the improvement of our manuscript. We have revised the manuscript according to the reviewers’ suggestions. All the changes are color-marked in the revised manuscript. The following are the detailed corrections and revisions point by point: 

(1) English needs to be further improved. Some sentences are not even completed, here are just a couple of examples:"However, little report on the effect of hydrogen on the corrosion behavior of metal materials in NH4Cl aqueous solution.""To study the effect of hydrogen on the corrosion behavior and passive film in NH4Cl solution, the potentiodynamic polarization curve and Electrochemical Impedance Spectroscopy (EIS) were performed."

Answer: Thanks for the reviewer’s comment. we have corrected the language carefully and hope this revised manuscript will be convincing.

 (2) What exactly are the composition and thickness of the passivation layer on the 321 SS? Can the author comment on it based on XPS measurements and/or literature reports?

Answer: Thanks for the reviewer’s comment. Firstly, the presence of passivation layer on the 321 SS can be seen in polarization curve and SEM result. Usually, the thickness of the passivation layer is only several nanometers and the depth of XPS test is about 10nm. So I consider that the XPS test result can reflect the change of passivation film. In this paper, it can be concluded that the outer of passivation film is composed of Fe3O4, Fe2O3 and FeOOH and the inner layer is Cr2O3 combined with the XPS and M-S test results. The explanation for this section can be seen in Part 2.4 in revised manuscript.

 (3) Based on the very comprehensive analyses in the manuscript, I agree with the authors that the passivation film loses its protection as the charging of hydrogen increases. This agrees with many other published reports. However, the authors state that the charging of hydrogen makes the passivation film thinner. I do not see direct evidence from the manuscript. Did the authors measure (or estimate) the thicknesses of the passivation layer before and after EHC? Or, do the authors just mean "pitting corrosion" instead? I might misunderstand the statement if the authors can clarify that would be great.

Answer: Thanks for the reviewer’s comment. The thicknesses of the passivation layer before and after EHC is not tested in the paper. Based on the analyses of Fig.6, Fig.7 and Fig.8, it can be obtained that hydrogen accelerates the anodic dissolution of the passive film. So we deduced that the hydrogen makes the passivation film thinner. This statement has been modified in the revised manuscript to avoid misunderstand.

Reviewer 3 Report

This reviewer is not expert in electro-chemistry and is therefore, with regret, not competent to assess this central aspect of the paper.

However, the aspect of defects and the phenomenon of hydrogen embrittlement in 321 stainless steel can be commented on. 

The blisters which are often seen on the surface of metals during the early stages of electro-chemical charging with hydrogen have recently been attributed to the large population of nanometer cracks (sometimes called bifilms) in stainless steels and Ni alloys which arise during the casting of the steel into ingots or continuous strands. The corrosion pits observed by the authors may be bifilm cracks lying parallel to the surface (the original random orientation of the bifilm cracks is rotated to be parallel to the axis of the bar by forging or rolling) from which the outer surface is gradually corroded away, leaving the inside surface of the crack visible.  The authors are invited to consider this model. 

Author Response

Thank you for your comments and suggestions on our manuscript which are very useful for the improvement of our manuscript. We have revised the manuscript according to the reviewers’ suggestions. All the changes are color-marked in the revised manuscript. The following are the detailed corrections and revisions point by point:

 The blisters which are often seen on the surface of metals during the early stages of electro-chemical charging with hydrogen have recently been attributed to the large population of nanometer cracks (sometimes called bifilms) in stainless steels and Ni alloys which arise during the casting of the steel into ingots or continuous strands. The corrosion pits observed by the authors may be bifilm cracks lying parallel to the surface (the original random orientation of the bifilm cracks is rotated to be parallel to the axis of the bar by forging or rolling) from which the outer surface is gradually corroded away, leaving the inside surface of the crack visible. The authors are invited to consider this model.

Answer: Thanks for the reviewer’s comment. I have been done similar experiment the reviewer mentioned before, seen in Reference [18]. Hydrogen cracks are preferentially produced at austenite grain boundaries, inclusions and the interface between δ ferrite and austenite under the condition of stress. This paper studies the effect of hydrogen on the corrosion resistance of 321 stainless steel in a non-stressed state. I think there is a difference between them. Of course, I will study this model the reviewer mentioned. Thanks again!

Round 2

Reviewer 1 Report

As authors have performed an adequate revise, the manuscript might be accepted for publication in the journal of Materials.